# Accelerometer-assessed outdoor physical activity is associated with meteorological conditions among older adults: Cross-sectional results from the OUTDOOR ACTIVE study

**Birte Marie Albrecht**⬚*, **Imke Stalling, Carina Recke, Karin Bammann**

Institute for Public Health and Nursing Sciences (IPP), University of Bremen, Bremen, Germany

* b.albrecht@uni-bremen.de

**Data Availability Statement:** All relevant data are within the manuscript and its Supporting Information files.

## Abstract

### Background

Meteorological conditions are potential determinants of physical activity (PA). A profound understanding of the determinants of PA behaviour is required for PA promotion. This study examined the association between accelerometer-assessed PA and meteorological conditions among older adults.

### Methods

This cross-sectional study included data of 577 adults aged 65–75 years living in Bremen, Germany (52% female; 3278 days). PA was measured with accelerometers for seven consecutive days (10/15-08/16). A threshold of 240 lx was used to differentiate between outdoor physical activity (OPA) and indoor physical activity (IPA). Linear mixed models estimated the association between PA (daily accelerometer counts per minute (CPM)) and meteorological factors (temperature, cloud cover, wind, and no precipitation) derived by principal component analysis.

### Results

The analyses showed associations between PA in CPM and the meteorological factors temperature (93.7; 95%-CL: 64.9, 122.5) and no precipitation (48.4; 95%-CL: 19.8, 77.0) in women and wind (-40.3; 95%-CL: -59.7, -20.8) and no precipitation (30.1; 95%-CL: 5.6, 54.6) in men. After distinguishing in OPA and IPA for a subsample of 128 participants (473 days), the sex differences were no longer present. OPA in CPM was associated with temperature (women: 174.5; 95%-CL: 81.3, 267.6; men: 183.3; 95%-CL: 81.2, 285.4), cloud cover (women: -153.0; 95%-CL: -200.3, -105.7; men: -123.2; 95%-CL: -174.7, -71.7), and wind (women: -118.6; 95%-CL: -189.6, -47.7; men: -96.9; 95%-CL: -177.0, -16.7). No association between OPA and no precipitation was found (women: 2.9; 95%-CL: -89.0, 94.8; men: -17.1; 95%-CL: -116.7, 82.4).

**Funding:** The OUTDOOR ACTIVE study is funded by the German Federal Ministry of Education and Research (BMBF; https://www.bmbf.de/; grant number: 01EL1422B). The funder had no role in study design, data collection and analysis, decision to publish, or preparation of the manuscript.

**Competing interests:** The authors have declared that no competing interests exist.

## Conclusions

The results of this study emphasize the importance of meteorological conditions as environmental determinants of PA among older adults. Meteorological conditions should be accounted for in the unbiased assessment of habitual PA and the development of PA promotion programs. Future research should focus on the associations of OPA and IPA with meteorological conditions in different climatic regions.

## Background

Regular physical activity (PA) is one of the key behavioural determinants of healthy ageing [1]. PA is positively associated with independent living, reduced disability, and improved quality of life [2]. Consequently, it is important to sustain a sufficient level of PA with increasing age. According to the World Health Organization older adults should accumulate at least 150 minutes of moderate-intensity PA throughout the week [3]. In Germany, however, the percentage of adults who meet these recommendations declines with increasing age: Only 18.0% of adults between 60 and 69 years and 13.6% of adults between 70 and 79 years engage in at least 150 minutes of moderate-intensity PA per week [4].

Evidence-based interventions in the promotion of PA require a profound understanding of the determinants of PA behaviour [5]. Ecological models suggest that intrapersonal, interpersonal, and environmental determinants are contributing to PA behaviour [6]. While intrapersonal and interpersonal variables are widely studied [5], research on environmental variables has only recently increased. Meteorological conditions count as potential environmental determinants of PA [7]. Walking is the most popular type of PA among adults over 65 years and being active outdoors is preferred [8]. Especially older adults, however, experience potential barriers to outdoor physical activity (OPA) with changing meteorological conditions. For example, slippery grounds due to rain and snow can increase the fear of falling and therefore prevent older adults from going outside [9]. In addition, the ability to thermoregulate deteriorates with increasing age. As a result, older adults have difficulties adapting to extreme temperatures and are therefore at a higher risk of hypo- or hyperthermia [10]. This further emphasizes the potential importance of meteorological conditions on PA behaviour of older adults. Next to the development of new PA promotion interventions, research regarding the association of meteorological conditions and PA is of additional value from a methodological viewpoint. The assessment of PA under different meteorological conditions in longitudinal studies and in the evaluation of PA promotion interventions can lead to wrong conclusions. For example, if PA is assessed in a period of unpleasant weather for baseline and a period of pleasant weather for follow-up the intervention effects could be overestimated. Therefore, an adjustment for meteorological conditions should be regarded.

Previous research indicates that there are statistically significant associations between accelerometer-assessed PA and different meteorological variables among older adults. Several studies showed a statistically significant increase in PA with rising temperature [9,11–14], decreasing precipitation [9,13,15,16], and longer days [11,13,17]. There is inconsistent evidence regarding the association between PA and wind speed [9,14]. None of these studies distinguished between OPA and indoor physical activity (IPA). Since meteorological conditions primarily influence the outdoor environment, more insight into the relationship is expected by looking at OPA and IPA separately. Timmermans et al. found a statistically significant increase in self-reported OPA with rising temperature and decreasing relative humidity [18].

Another study used GPS data to estimate the time walked and cycled by older adults. Their results indicate a positive association between walking and cycling time and temperature. In addition, walking time was positively associated with wind speed and negatively associated with precipitation [19]. To the best of the authors' knowledge, no study to date has focused on the association between accelerometer-assessed OPA and meteorological conditions among older adults.

This study aims to investigate the relationship between PA and meteorological conditions among older adults. In addition, this study differentiates between OPA and IPA for a subsample and explores their respective associations with meteorological conditions.

## Methods

### Study design and population

The OUTDOOR ACTIVE study is part of the regional prevention network AEQUIPA and aims to develop and implement a community-based OPA promotion program in older adults [20]. Eligible for OUTDOOR ACTIVE were all non-institutionalised adults between the age of 65 and 75 years residing in the district Hemelingen in the city of Bremen, located in North-Western Germany. Address data were obtained in August 2015 from the registry office of Bremen. Eligible individuals were initially contacted by letter, followed by a phone contact if the number was listed in one of the available registers. All participants provided written informed consent. The study was approved by the ethical committee of the University of Bremen.

The OUTDOOR ACTIVE study includes a baseline and follow-up assessment, of which only the data collected at baseline were used in the present paper. Baseline assessment took place between October 2015 and August 2016 and consisted of 1) a self-administered questionnaire focusing on intrapersonal, interpersonal, and environmental determinants of PA, 2) a short health examination consisting of physical examination (anthropometry and blood pressure) and fitness test (Senior Fitness Test [21] and handgrip strength test), and 3) a seven day accelerometer-measurement of PA.

### Measures

**Accelerometer-assessed physical activity.** Participants were asked to wear an ActiGraph wGT3X-BT accelerometer (ActiGraph LLC, Pensacola, FL, USA) for seven consecutive days (24 h) on their non-dominant wrist. Epoch length was set to 30 Hz. Accelerometer data were downloaded using ActiLife (Version 6.13.3, ActiGraph LLC, Pensacola, FL, USA) and prepared for the statistical analyses in RStudio (Version 1.0.136, RStudio Inc., Boston, MA, USA). First and last wear days were excluded from the analyses. Non-wear time was defined as 30 minutes with zero counts and only days with a wear time of at least 20 hours counted as valid. Participants were excluded from the analyses if they had not at least one valid day of accelerometer data. Daily average accelerometer vector magnitude counts per minute (CPM) were included in the analyses as the outcome variable. For the OPA/IPA-stratified analyses the integrated light sensor of the accelerometer was used to differentiate between OPA and IPA. As proposed by the literature, values of at least 240 lx were categorised as outdoor environment. Values below 240 lx were defined as indoor environment [22]. Daily average CPM of OPA and IPA were calculated.

**Meteorological variables.** Meteorological data were retrieved online from the German Weather Service. The weather station is located at Bremen airport. All participants lived within approximately 11 km maximum distance from the weather station. Available daily data for Bremen consisted of mean temperature (in ˚C), minimum temperature at 2 m (in ˚C), minimum temperature at 5 cm (in ˚C), mean vapor pressure (in hPa), maximum temperature at 2

m (in˚C), sunshine (in h), mean relative humidity (in %), mean cloud cover (in 1/8), mean wind speed (in m/s), maximum wind speed (in m/s), mean air pressure (in hPa), snow depth (in cm), and total precipitation (in mm) [23]. In addition, day length (in h) was calculated as the time between sunrise and sunset [24].

The city of Bremen lies within the temperate climate zone [25]. During the observation period, the coldest month was January with a mean temperature of 1.5˚C (10th percentile: -5.4˚C, 90th percentile: 7.1˚C) and the warmest month was July with a mean temperature of 18.5˚C (10th percentile: 14.9˚C, 90th percentile: 22.6˚C). Mean sunshine varied from 1.5 h per day (10th percentile: 0.0 h, 90th percentile: 4.8 h) in January to 7.8 h per day (10th percentile: 1.0 h, 90th percentile: 14.5 h) in May. Mean wind speed was lowest in October with 3.0 m/s (10th percentile: 1.5 m/s, 90th percentile: 4.6 m/s) and highest in November (10th percentile: 1.9 m/s, 90th percentile: 8.8 m/s) and December (10th percentile: 3.3 m/s, 90th percentile: 7.9 m/s) with 5.5 m/s. Mean precipitation was lowest in August (mean: 0.6 mm; 10th percentile: 0.0 mm, 90th percentile: 1.9 mm) and highest in June (mean: 3.5 mm; 10th percentile: 0.0 mm, 90th percentile: 13.2 mm) (Fig 1).

**Demographic and anthropometric information.** Information on participant's sex, educational status, self-rated health, number of chronic diseases, and number of daily taken medications was assessed through a self-administered questionnaire. Educational status was

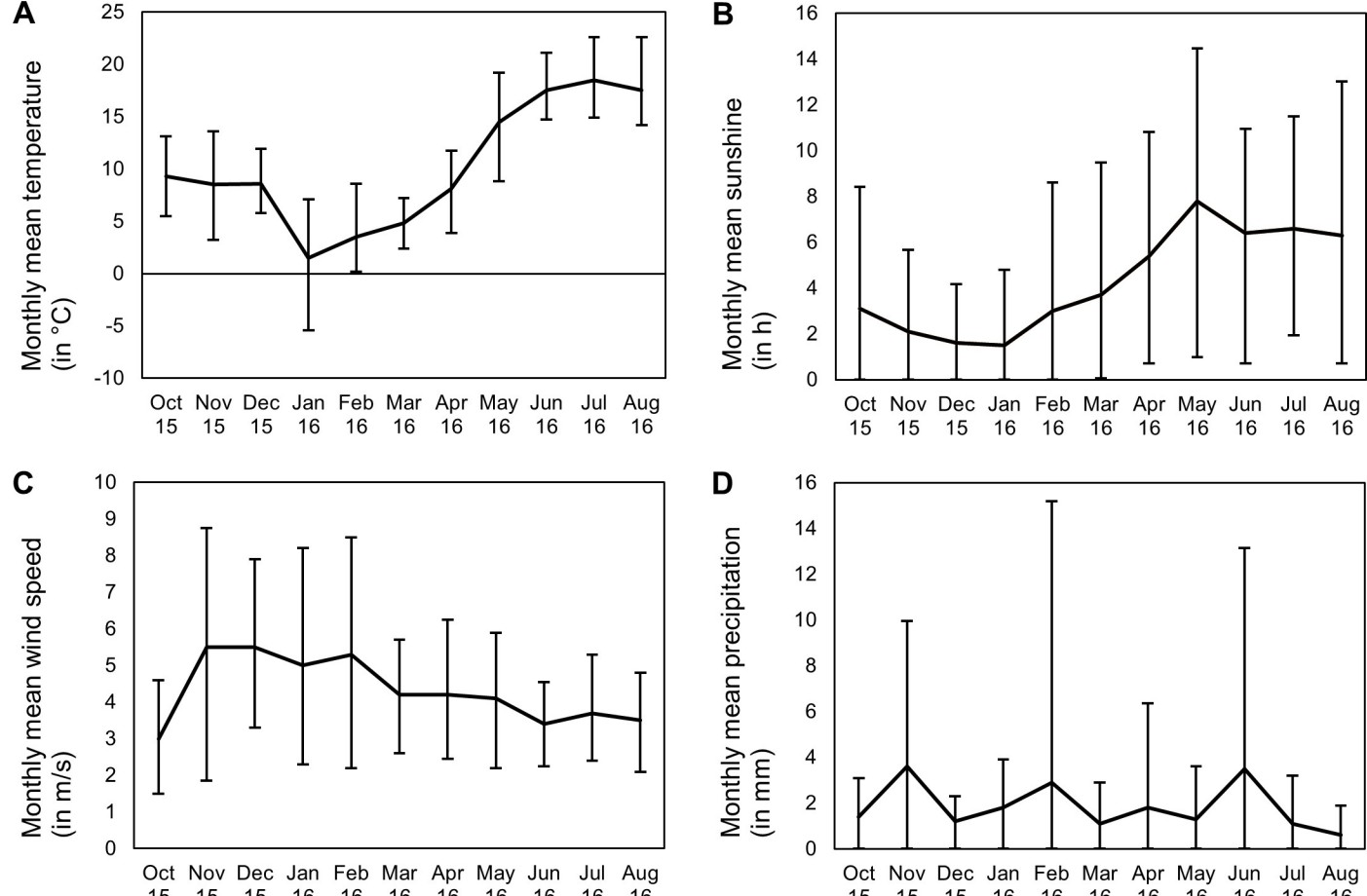

**Fig 1. Monthly means and their 10th and 90th percentile of meteorological conditions during the observation period.** (A) Temperature. (B) Sunshine. (C) Wind speed. (D) Precipitation.

classified into six categories according to the International Standard Classification of Education 1997 [26]. Self-rated health was assessed with a single item from the SF-36 questionnaire [27]. Data on age, height, and body weight were collected as part of the health examination. Height was measured with a Seca 217 mobile stadiometer (Seca GmbH & Co. KG, Hamburg, Germany) and body weight with a Kern MPC 250K100M personal floor scale (Kern & Sohn GmbH, Ballingen, Germany). Body mass index (BMI) was calculated as the quotient of body weight (in kg) and the squared height (in m).

## Statistical analyses

Absolute and relative frequencies were calculated for educational status, BMI, self-rated health, number of chronic diseases, and number of daily taken medications. Means and standard deviations were determined for age and PA. All descriptive analyses of the individual variables were done for the total population and for women and men separately. Monthly means and the 10th and 90th percentile were calculated for all meteorological conditions during the observation period.

Correlation coefficients $\geq 0.8$ between several meteorological conditions indicated multicollinearity (S1 Table). Therefore, a principal component analysis (PCA) was conducted. All meteorological variables were included as continuous variables. The Kaiser-Meyer-Olkin measure of sampling adequacy was 0.75 and Bartlett's test for sphericity resulted in a $\chi^2$ value of 6141.0 (df = 91, p<0.01). Both tests confirmed the appropriateness of data to conduct a PCA [28]. Varimax rotation was applied to achieve a better allocation of the variables to one factor. Eigen values $\geq 1.0$ were used to determine the number of relevant factors [29]. Four meteorological factors were identified: 1) temperature, 2) cloud cover, 3) wind, and 4) no precipitation. These four factors explained 81.9% of the total variance in meteorological conditions (Table 1). For each relevant factor, daily values were included as independent variables in the model.

**Table 1. Factor loadings of meteorological variables and explained variance of factors derived by principal component analysis (n = 296).**

| | Factor 1 Temperature | Factor 2 Cloud cover | Factor 3 Wind | Factor 4 No precipitation |
|---|---|---|---|---|
| Mean temperature (˚C) | 0.97 | -0.20 | 0.01 | 0.08 |
| Minimum temperature at 2 m (˚C) | 0.97 | 0.00 | 0.08 | 0.07 |
| Minimum temperature at 5 cm (˚C) | 0.97 | 0.05 | 0.09 | 0.04 |
| Mean vapor pressure (hPa) | 0.96 | 0.07 | -0.07 | 0.02 |
| Maximum temperature at 2 m (˚C) | 0.92 | -0.33 | -0.04 | 0.06 |
| Day length (h) | 0.68 | -0.46 | -0.27 | -0.19 |
| Sunshine (h) | 0.22 | -0.90 | -0.13 | 0.09 |
| Mean relative humidity (%) | -0.27 | 0.86 | -0.15 | 0.02 |
| Mean cloud clover (1/8) | 0.14 | 0.83 | 0.06 | -0.22 |
| Mean wind speed (km/h) | -0.10 | 0.05 | 0.95 | -0.10 |
| Maximum wind speed (km/h) | 0.06 | 0.00 | 0.94 | -0.14 |
| Mean air pressure (hPa) | -0.01 | -0.11 | -0.27 | 0.79 |
| Snow depth (cm) | -0.38 | 0.00 | -0.13 | -0.58 |
| Precipitation (mm) | 0.16 | 0.33 | 0.27 | -0.43 |
| Variance explained | 0.385 | 0.195 | 0.148 | 0.091 |

Factor loadings > 0.4 (absolute values) are shown in bold characters.

Total variance explained: 0.819

Linear mixed models were fitted to account for the repeated-measures structure of data. Days and study subjects were included as random factors. All models were stratified by sex and covariates were selected based on the literature [5]. First, unadjusted associations between the meteorological factors and PA in CPM were estimated, followed by adjusting for age and BMI. Participants with missing data of covariates were excluded from the adjusted analyses. The equation of the adjusted linear mixed model took the following form:

$$Y_{ij}(PA\ in\ CPM)$$
$$= (\gamma_0 + u_{ij}) + \gamma_1 \times temperature_{ij} + \gamma_2 \times cloud\ cover_{ij} + \gamma_3 \times wind_{ij} + \gamma_4 \times no\ precipitation_{ij}$$
$$+ \gamma_5 \times age_{ij} + \gamma_6 \times BMI_{ij} + \varepsilon_{ij}$$

Y = outcome variable
γ = mean estimate for the parameter
u = random effect
i = study subject
j = day
ε = residual

Second, the same models were estimated with PA in CPM stratified into OPA and IPA. Due to the location of the accelerometer at the wrist, the possibility of the light sensor being covered by clothing cannot be ruled out. Therefore, only days reaching a maximum temperature of at least 20˚C were included in the OPA/IPA-stratified analyses. Furthermore, a sensitivity analysis with a threshold of 500 lx (indoor environment < 500 lx; outdoor environment ≥ 500 lx) was conducted.

All statistical analyses were performed with SAS® University Edition (SAS Institute Inc., Cary, NC, USA).

## Results

Of the 4304 potentially eligible individuals, 615 individuals were not able to participate in the study due to acute health problems (n = 242) or death (n = 56), language barriers (n = 22) or because they moved outside of the survey region (n = 295). 720 of the 3689 confirmed eligible individuals were never reached; 2052 individuals refused to participate. 916 individuals participated in at least one part of the study and of those, 577 participants were included in the cross-sectional analyses.

Table 2 shows descriptive characteristics of the study population. The mean age was 69.5± 2.9 years and 52.0% were female. More than one third of the population (34.6%) reached an ISCED level ≥ 5. Most participants had normal weight (30.6%) or were overweight (44.0%). Overall, 58.8% of participants described their health status as good and 21.5% as very good or excellent. Approximately one quarter of the study population declared no chronic diseases (27.9%) and no daily medication intake (24.3%). Participants provided 3278 valid days of accelerometer data. The mean number of valid accelerometer days per person was 5.7±0.8 with no differences by sex. The mean of daily CPM recorded was 1628.1±505.8. Women (1784.6±519.9 CPM) accumulated more accelerometer CPM than men (1457.7±429.6 CPM).

Table 3 reports the results of the main analyses regarding the association between the four meteorological factors and PA in CPM stratified by sex. The results of the linear mixed models remained similar after adjusting for age and BMI. The main analyses showed differences between women and men. The adjusted models indicated a statistically significant positive association between PA and temperature in women (93.7; 95%-CL: 64.9, 122.5). There was no statistically significant relationship between PA and temperature in men (14.4; 95%-CL: -9.0, 37.8).

**Table 2. Characteristics of the study population.**

| | Total (n = 577) | Women (n = 300) | Men (n = 277) |
|---|---|---|---|
| | n (%) | n (%) | n (%) |
| Education | | | |
| Basic education (ISCED level 1 + 2) | 90 (16.7) | 74 (26.1) | 16 (6.2) |
| Specialized education (ISCED level 3 + 4) | 263 (48.7) | 155 (54.8) | 108 (42.0) |
| Advanced education (ISCED level $\geq$ 5) | 187 (34.6) | 54 (19.1) | 133 (51.8) |
| Body mass index (kg/m$^2$) | | | |
| Underweight ($<$ 18.5) | 4 (0.7) | 4 (1.3) | 0 |
| Normal weight (18.5 - $<$ 25) | 176 (30.6) | 108 (36.2) | 68 (24.6) |
| Overweight (25 - $<$ 30) | 253 (44.0) | 106 (35.6) | 147 (53.1) |
| Obesity ($\geq$ 30) | 142 (24.7) | 80 (26.9) | 62 (22.4) |
| Self-rated health | | | |
| Less good or bad | 106 (19.7) | 64 (22.9) | 42 (16.2) |
| Good | 317 (58.8) | 163 (58.4) | 154 (59.2) |
| Very good or excellent | 116 (21.5) | 52 (18.6) | 64 (24.6) |
| Number of chronic diseases | | | |
| None | 150 (27.9) | 53 (18.8) | 97 (37.9) |
| 1 | 188 (34.9) | 101 (35.8) | 87 (34.0) |
| 2 | 117 (21.7) | 74 (26.2) | 43 (16.8) |
| 3 | 52 (9.7) | 32 (11.3) | 20 (7.8) |
| $\geq$ 4 | 31 (5.8) | 22 (7.8) | 9 (3.5) |
| Number of daily taken medications | | | |
| None | 123 (24.3) | 52 (19.7) | 71 (29.2) |
| 1 | 124 (24.5) | 66 (25.0) | 58 (23.9) |
| 2 | 85 (16.8) | 55 (20.8) | 30 (12.3) |
| 3 | 50 (9.9) | 28 (10.6) | 22 (9.1) |
| 4 | 49 (9.7) | 22 (8.3) | 27 (11.1) |
| $\geq$ 5 | 76 (15.0) | 41 (15.5) | 35 (14.4) |
| | n | n | n |
| Observed accelerometer days | 3278 | 1709 | 1569 |
| | Mean (SD) | Mean (SD) | Mean (SD) |
| Age (years) | 69.5 (2.9) | 69.6 (2.9) | 69.3 (2.8) |
| Physical activity (average accelerometer CPM) | 1628.1 (505.8) | 1784.6 (519.9) | 1457.7 (429.6) |
| Valid accelerometer days | 5.7 (0.8) | 5.7 (0.7) | 5.6 (0.9) |

ISCED: International Standard Classification of Education

SD: Standard deviation

CPM: Counts per minute

PA was not associated with cloud cover in women (-22.8; 95%-CL: -46.8, 1.3) and men (-10.8; 95%-CL: -31.1, 9.6). While men accumulated statistically significantly less PA with increasing wind (-40.3; 95%-CL: -59.7, -20.8), there was no statistically significant association in women (8.6; 95%-CL: -14.2, 31.3). The results indicated a positive relationship between PA and no precipitation in women (48.4; 95%-CL: 19.8, 77.0) and men (30.1; 95%-CL: 5.6, 54.6). PA was negatively associated with increasing age (women: -36.4; 95%-CL: -44.4, -28.3; men: -21.2; 95%-CL: -28.6, -13.8) and BMI (women: -18.9; 95%-CL: -24.0, -13.8; men: -24.8; 95%-CL: -30.3, -19.3).

Tables 4 and 5 present the results of the association between OPA and IPA in CPM with the meteorological factors. Overall, 473 days of PA assessment from 68 women and 60 men were

**Table 3. Association of PA (average accelerometer CPM) and meteorological factors.**

| | PA (unadjusted) | | PA (adjusted) | |
|---|---|---|---|---|
| | Women (n = 300, 1709 days) | Men (n = 277, 1569 days) | Women (n = 298, 1698 days) | Men (n = 277, 1569 days) |
| | β (95%-CL) | β (95%***-CL) | β (95%-CL) | β (95%-CL) |
| Factor 1 Temperature | 89.0 (59.3, 118.7)*** | 5.7 (-18.5, 29.8) | 93.7 (64.9, 122.5)*** | 14.4 (-9.0, 37.8) |
| Factor 2 Cloud cover | -26.0 (-50.7, -1.3)* | -1.2 (-22.0, 19.7) | -22.8 (-46.8, 1.3) | -10.8 (-31.1, 9.6) |
| Factor 3 Wind | 14.7 (-8.7, 38.0) | -34.7 (-54.8, -14.7)*** | 8.6 (-14.2, 31.3) | -40.3 (-59.7, -20.8)*** |
| Factor 4 No precipitation | 45.2 (15.8, 74.6)** | 28.0 (2.8, 53.2)* | 48.4 (19.8, 77.0)*** | 30.1 (5.6, 54.6)* |
| Age (years) | | | -36.4 (-44.4, -28.3)*** | -21.2 (-28.6, -13.8)*** |
| Body mass index (kg/m$^2$) | | | -18.9 (-24.0, -13.8)*** | -24.8 (-30.3, -19.3)*** |

* $p$-value < 0.05

** $p$-value < 0.01

*** $p$-value < 0.001

PA: Physical activity

CPM: Counts per minute

CL: Confidence limits

Linear mixed models (random factors: days, study subjects), stratified by sex, unadjusted and adjusted for age and body mass index.

Meteorological factors were derived by principal component analysis.

included in the OPA/IPA-stratified analyses. The unadjusted and adjusted models indicated the same statistically significant associations between OPA and IPA and the meteorological factors. In contrast to the main analyses, the results of the OPA/IPA-stratified analyses showed no distinct differences of women and men. The adjusted linear mixed models indicated a significant increase in OPA with rising temperature (women: 174.5; 95%-CL: 81.3, 267.6; men: 183.3; 95%-CL: 81.2, 285.4), decreasing cloud cover (women: -153.0; 95%-CL: -200.3, -105.7; men: -123.2; 95%-CL: -174.7, -71.7), and decreasing wind (women: -118.6; 95%-CL: -189.6; -47.7; men: -96.9; 95%-CL: -177.0, -16.7). There was no statistically significant association of OPA and no precipitation (women: 2.9; 95%-CL: -89.0, 94.8; men: -17.1; 95%-CL: -116.7, 82.4). In women and men, no statistically significant relationship between IPA and temperature (women: 77.7; 95%-CL: -58.7, 214.1; men: 72.0; 95%-CL: -67.0, 151.0) or wind (women: 48.1; 95%-CL: -55.7, 152.0; men: -78.7; 95%-CL: -164.3, 6.9) could be shown. Cloud cover was associated with an increase in IPA in women (98.0; 95%-CL: 28.8, 167.3) and men (74.1; 95%-CL: 19.1, 129.1). IPA was positively associated with no precipitation in women (204.7; 95%-CL: 70.0, 339.3), but not in men (-7.3; 95%-CL: -113.5, 99.0). The sensitivity analyses with a threshold of 500 lx revealed similar results (S2 and S3 Table).

## Discussion

The study showed sex-specific associations between the amount of accelerometer-assessed PA and the meteorological factors. PA was associated with temperature and precipitation in women, while men showed an association of PA with wind and precipitation. After distinguishing in OPA and IPA the sex differences were no longer present. OPA was associated with

**Table 4. Association of OPA (average accelerometer CPM) and meteorological factors.**

| | OPA (unadjusted) | | OPA (adjusted) | |
|---|---|---|---|---|
| | Women (n = 68, 238 days) | Men (n = 60, 235 days) | Women (n = 68, 238 days) | Men (n = 60, 235 days) |
| | β (95%-CL) | β (95%-CL) | β (95%-CL) | β (95%-CL) |
| Factor 1 Temperature | 167.4 (72.4, 262.3)*** | 186.1 (84.2, 288.0)*** | 174.5 (81.3, 267.6)*** | 183.3 (81.2, 285.4)*** |
| Factor 2 Cloud cover | -147.1 (-194.7, -99.4)*** | -122.3 (-174.0, -70.7)*** | -153.0 (-200.3, -105.7)*** | -123.2 (-174.7, -71.7)*** |
| Factor 3 Wind | -104.5 (-176.4, -32.6)** | -99.9 (-180.1, -19.8)* | -118.6 (-189.6, -47.7)** | -96.9 (-177.0, -16.7)* |
| Factor 4 No precipitation | 14.5 (-79.6, 108.5) | -15.1 (-114.6, 84.4) | 2.9 (-89.0, 94.8) | -17.1 (-116.7, 82.4) |
| Age (years) | | | -19.3 (-30.7, -7.9)** | -14.5 (-28.7, -0.3)* |
| Body mass index (kg/m$^2$) | | | -11.6 (-20.7, -2.5)* | -0.7 (-11.7, 10.2) |

* $p$-value < 0.05

** $p$-value < 0.01

*** $p$-value < 0.001

OPA: Outdoor physical activity

CPM: Counts per minute

CL: Confidence limits

Linear mixed models (random factors: days, study subjects), stratified by sex, unadjusted and adjusted for age and body mass index.

Meteorological factors were derived by principal component analysis.

Includes only days with a maximum temperature ≥ 20˚C.

OPA defined as lx ≥ 240.

temperature, cloud cover, and wind. OPA was negatively and IPA was positively associated with cloud cover. No relationship between OPA and precipitation was found.

This study showed a positive association between the amount of total PA and temperature in women but not in men. This is in contrast to the study results of Aspvik et al. in Norway and Klenk et al. in Germany, as they reported statistically significant associations for women and men [9,14]. Other studies found a positive association between PA and temperature as well. They, however, did not report results stratified by sex [11–13]. The negative association between PA and precipitation is in accordance with prior studies [9,13,15,16]. Precipitation is not only unpleasant, it might also increase the fear of falling due to slippery grounds [9]. As previously stated, the evidence regarding the relationship of PA and wind is conflicting. The reported negative association of PA and wind in men is in line with the results of Klenk et al. Their analyses, however, also indicated a reduction in PA with increasing wind in women [9]. In contrast, Aspvik et al. found a statistically significant positive association of PA and wind in women but not in men [14].

The study showed distinct sex differences regarding the association between total PA and meteorological conditions. These analyses, however, did not account for the different PA behaviours of women and men. Previous studies reported that recreational and occupational activities with a moderate to vigorous intensity are more prevalent in men than in women. At the same time, women accumulate more low-intensity PA by performing tasks around the household [30–32]. This results in women spending more time indoors than men [31]. Therefore, men accumulate more PA outdoors while women tend to be active indoors. Environmental conditions primarily influence the outdoor environment. The inclusion of total PA,

**Table 5. Association of IPA (average accelerometer CPM) and meteorological factors.**

| | IPA (unadjusted) | | IPA (adjusted) | |
|---|---|---|---|---|
| | Women (n = 68, 238 days) | Men (n = 60, 235 days) | Women (n = 68, 238 days) | Men (n = 60, 235 days) |
| | β (95%-CL) | β (95%-CL) | β (95%-CL) | β (95%-CL) |
| Factor 1 Temperature | 78.3 (-64.8, 221.4) | 39.2 (-69.2, 147.6) | 77.7 (-58.7, 214.1) | 42.0 (-67.0, 151.0) |
| Factor 2 Cloud cover | 98.2 (26.4, 170.0)** | 76.1 (21.2, 131.1)** | 98.0 (28.8, 167.3)** | 74.1 (19.1, 129.1)** |
| Factor 3 Wind | 63.2 (-45.1, 171.4) | -77.6 (-162.8, 7.6) | 48.1 (-55.7, 152.0) | -78.7 (-164.3, 6.9) |
| Factor 4 No precipitation | 234.9 (93.2, 376.6)** | -10.0 (-115.8, 95.8) | 204.7 (70.0, 339.3)** | -7.3 (-113.5, 99.0) |
| Age (years) | | | -44.6 (-61.2, -27.9)*** | -10.5 (-25.7, 4.7) |
| Body mass index (kg/m$^2$) | | | -13.6 (-26.9, -0.2)* | -5.2 (-16.9, 6.5) |

* $p$-value < 0.05

** $p$-value < 0.01

*** $p$-value < 0.001

IPA: Indoor physical activity

CPM: Counts per minute

CL: Confidence limits

Linear mixed models (random factors: days, study subjects), stratified by sex, unadjusted and adjusted for age and body mass index.

Meteorological factors were derived by principal component analysis.

Includes only days with a maximum temperature ≥ 20˚C.

IPA defined as lx < 240.

instead of OPA and IPA, as outcome variable can result in the wrong conclusions. The results of the OPA/IPA-stratified analyses confirmed this assumption, since they no longer showed distinct sex differences.

Even though only relatively warm days with a maximum temperature of at least 20˚C were included in the OPA/IPA-stratified analyses, the results revealed an increase in OPA with rising temperatures. This is in line with previous studies examining OPA [18,19]. In contrast, the results of other studies showed a peak in PA at a certain temperature, after which a decrease of PA was seen. Togo et al. proposed a peak in step counts at a temperature of 17˚C [33], while the results of Brandon et al. indicated a peak in PA in CPM at 20˚C [34]. It must be noted, that these studies did not examine OPA. OPA was negatively and IPA was positively associated with cloud cover (S1 Fig). A possible explanation is that either part of OPA is substituted with IPA when it is cloudy or that part of IPA is substituted with OPA when it is sunny. Price et al. found an increased use of trails among older adults when it was sunny. As they solely examined trail use, it is unclear whether a substitution of IPA took place [35]. Further research is needed to understand this finding. The results regarding the association of OPA and IPA with precipitation must be interpreted carefully as only approximately 20% of days included in the subsample had precipitation. In contrast to the results of the main analyses, no association between OPA and precipitation was found. This is in line with the results of Timmermans et al. [18]. In contrast, Prins et al. found a negative association between walking time and precipitation. But it must be noted, that they not specifically assessed OPA but GPS-measured walking time [19].

Even though meteorological conditions cannot be modified, these results can be important for future research. New PA promotion concepts for older adults should account for meteorological conditions. It is necessary to develop interventions to reduce the negative associations between PA and meteorological conditions. One approach is the identification of personal attributes that moderate the negative relationship. Hoppmann et al. identified PA intentions as a potential moderating variable in the association of PA and precipitation [15]. As PA intentions are potentially modifiable, this finding provides an approach for new PA promotion programs [36]. An alternative approach could be to encourage IPA during adverse meteorological conditions to substitute the decrease in OPA [37]. For this, it is required to provide easily accessible indoor leisure facilities [38]. Further research should focus on the differentiation between OPA and IPA and their respective associations with meteorological conditions. This promises more insight into the exact associations and could help in the development of new intervention strategies.

In addition, this study is of value from a methodological viewpoint. The results indicate that meteorological conditions should be accounted for in the assessment of PA under different meteorological conditions. The use of factors derived by PCA can help to adjust for several highly correlated meteorological variables at the same time.

As there are several climate zones with different meteorological conditions, the results cannot be generalised to other parts of the world or to other years where the weather conditions are different. This study was conducted in a temperate climate with a relatively mild winter and summer. Different results are expected in regions with more extreme meteorological conditions. This is especially the case for the results of the OPA/IPA-stratified analyses, as only relatively warm days with a maximum temperature of at least 20˚C were included. Of those, only approximately 20% had precipitation. Future research should focus on the assessment of OPA and IPA in different climatic regions, and over a longer period.

## Strengths and limitations

This study has some limitations, which should be addressed by further research. Each participant wore the accelerometer for only seven days. Therefore, every participant experienced different meteorological conditions. A longer observation period per participant would be desirable. However, it seems unlikely that the results are biased by the data structure, as participant and day were included as random effects in the regression equations and, as invitations were sent out in a random pattern, recruitment should ensure that any participant characteristics are independent from date of data collection.

Because the accelerometer has no GPS, the exact position of the participant is unknown. It is possible that some participants travelled further away and experienced different meteorological conditions. The participants had to be in Bremen to receive and return the accelerometer within the following week. While some participants were probably not permanently in Bremen during accelerometer data collection, we have no reason to assume that this happened to a larger scale in the sample. Therefore, the risk of exposure misclassification is low.

The results of the OPA/IPA-stratified analyses must be interpreted with caution. Even though only days with a maximum temperature of at least 20˚C were included in the analyses, it cannot be guaranteed that the light sensor of the accelerometer was not covered by clothing. An underestimation of the ambient light is likely, which causes a misclassification of OPA and IPA. In addition, not a lot of research has been done regarding the lux threshold to differentiate between indoor and outdoor environment. A sensitivity analysis with a higher threshold, however, revealed similar results. The large confidence intervals indicated that the subsample was relatively small. This was especially the case for the results of precipitation, as only approximately 20% of days with a maximum temperature of at least 20˚C had precipitation.

One of the strengths of this study is the homogeneity of the study population. All participants resided in Bremen's district Hemelingen. Therefore, data was already controlled for other determinants, such as the built environment. The analyses included solely objective data for exposure and outcome. Accelerometers are considered to be a reliable and valid tool in the PA measurement of older adults [39]. In addition, meteorological variables were included as factors in the analyses to avoid loss of information while accounting for the high level of multicollinearity in the meteorological data. To the best of the authors' knowledge, this is the first study to investigate the association between objectively differentiated accelerometer-assessed OPA and IPA and meteorological conditions. Our approach allowed for more insight into the exact associations of PA and meteorological conditions.

## Conclusions

In conclusion, the findings of this study emphasize the importance of meteorological conditions as an environmental determinant of PA among older adults. Therefore, they should be regarded in the assessment of PA and the development of PA promotion programs. Distinguishing between OPA and IPA is necessary to account for different PA behaviours of women and men. Further research should differentiate between OPA and IPA to obtain a better understanding of the relationship with meteorological conditions in different climatic regions.

## Supporting information

**S1 Fig. Outdoor and indoor physical activity (average accelerometer counts per minute) by factor 2: cloud cover (n = 473).**
(TIF)

**S1 Table. Pearson correlation coefficients between all meteorological variables.**
(PDF)

**S2 Table. Association of outdoor physical activity (average accelerometer counts per minute) and meteorological factors with outdoor physical activity defined as lx $\geq$ 500.**
(PDF)

**S3 Table. Association of indoor physical activity (average accelerometer counts per minute) and meteorological factors with indoor physical activity defined as lx $<$ 500.**
(PDF)

## Acknowledgments

The authors would like to thank all participants of the study and the German Weather Service for supplying the meteorological data.

## Author Contributions

**Conceptualization:** Birte Marie Albrecht, Karin Bammann.

**Data curation:** Birte Marie Albrecht.

**Formal analysis:** Birte Marie Albrecht.

**Funding acquisition:** Karin Bammann.

**Investigation:** Carina Recke, Karin Bammann.

**Methodology:** Birte Marie Albrecht.

**Project administration:** Karin Bammann.

**Supervision:** Karin Bammann.

**Validation:** Imke Stalling.

**Visualization:** Birte Marie Albrecht.

**Writing – original draft:** Birte Marie Albrecht.

**Writing – review & editing:** Imke Stalling, Carina Recke, Karin Bammann.

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
