## [Decision Letter · Decision Letter 0]

21 Nov 2019

PONE-D-19-20850

Accelerometer-assessed outdoor physical activity is associated with meteorological conditions among older adults: cross-sectional results from the OUTDOOR ACTIVE study

PLOS ONE

Dear Mrs. Albrecht,

Thank you for submitting your manuscript to PLOS ONE. After careful consideration, we feel that it has merit but does not fully meet PLOS ONE’s publication criteria as it currently stands. Therefore, we invite you to submit a revised version of the manuscript that addresses the points raised during the review process.

Overall the manuscript is well written however some key missing details within the methodology and results need to be addressed/added. The comments and suggestion would serve to make this much stronger paper

We would appreciate receiving your revised manuscript by Jan 05 2020 11:59PM. To enhance the reproducibility of your results, we recommend that if applicable you deposit your laboratory protocols in protocols.io, where a protocol can be assigned its own identifier (DOI) such that it can be cited independently in the future. For instructions see: http://journals.plos.org/plosone/s/submission-guidelines#loc-laboratory-protocols

We look forward to receiving your revised manuscript.

Kind regards,

Joe Robert Nocera

Academic Editor

PLOS ONE

Journal Requirements:

Reviewers' comments:

Reviewer's Responses to Questions

**Comments to the Author**

1. Is the manuscript technically sound, and do the data support the conclusions?

Reviewer #1: Yes

2. Has the statistical analysis been performed appropriately and rigorously? 

Reviewer #1: Yes

3. Have the authors made all data underlying the findings in their manuscript fully available?

Reviewer #1: Yes

4. Is the manuscript presented in an intelligible fashion and written in standard English?

Reviewer #1: Yes

5. Review Comments to the Author

Reviewer #1: In this cross-sectional study, the authors examined the association between physical activity levels (indoor and outdoor) and meteorological conditions. Overall, the manuscript was very well written and clearly organized. The methodology was fairly sound, though I do have some questions.

Comments by section:

BACKGROUND:

Lines 60-61: This sentence needs a reference.

Lines 64-82: I understand what the authors are saying, but an example would be helpful here. For example, if PA was assessed during a period of pleasant weather, this may make it appear that individuals are much more active than they usually area (factoring in multiple weeks of mixed weather).

METHODS:

Lines 106-108: Why the max age limit of 75 years?

Accelerometer assessment of activity:

Lines 127-128: Non-wear time in older adults is typically set at 60 minutes (or 90 minutes for institutionalized older adults). Why was 30 minutes set? Also, the authors specify that a valid day was considered as 20 hours of wear (line 128). Were the participants were sleeping with the devices on? Please clarify. Normally, 10 hours is considered the cut-off for minimum wear time to equal a valid day.

The authors state that the minimum was one day of wear time. How can you address intra-person variability in PA by climate with just 1 day of measurement?

Please explain why activity level was classified based on mean daily CPM rather than CPM thresholds?

RESULTS:

Line 228 and in table 1: Include the mean # of valid days per person (with range) here and in table 2.

Lines 244-247: Was the relationship between PA and age stronger in women or men - or no difference?

DISCUSSION:

Lines 311-314: What degree of substitution takes place? Do individuals engage in equal amounts of IPA when weather makes OPA untenable (or vice versa)? Or is there magnitude of change? (I.e. said individual engages in half the IPA that they would have if weather was nicer and they went outside?)

6. PLOS authors have the option to publish the peer review history of their article (what does this mean?). If published, this will include your full peer review and any attached files.

Reviewer #1: No

---

## [Author Response · Author response to Decision Letter 0]

19 Dec 2019

Dear Dr. Nocera,

we are sending the revised manuscript 

Accelerometer-assessed outdoor physical activity is associated with meteorological conditions among older adults: cross-sectional results from the OUTDOOR ACTIVE study.

We feel that the paper greatly improved by the changes requested by the reviewer and are very grateful for the comments. Please find below a point-by-point reply to the comments. 

Kind regards

Birte Albrecht, also on behalf of the co-authors.

Reviewer 1:

In this cross-sectional study, the authors examined the association between physical activity levels (indoor and outdoor) and meteorological conditions. Overall, the manuscript was very well written and clearly organized. The methodology was fairly sound, though I do have some questions.

Comments by section:

BACKGROUND:

Lines 60-61: This sentence needs a reference.

Reply: All three sentences refered to the same reference. We changed the text to clarify this (lines 59-63).

Lines 64-82: I understand what the authors are saying, but an example would be helpful here. For example, if PA was assessed during a period of pleasant weather, this may make it appear that individuals are much more active than they usually area (factoring in multiple weeks of mixed weather).

Reply: An example was added in lines 80-83.

METHODS:

Lines 106-108: Why the max age limit of 75 years?

Reply: The main research question of the OUTDOOR ACTIVE study focuses on PA behaviour changes around retirement, hence the narrow age range.

Accelerometer assessment of activity:

Lines 127-128: Non-wear time in older adults is typically set at 60 minutes (or 90 minutes for institutionalized older adults). Why was 30 minutes set? Also, the authors specify that a valid day was considered as 20 hours of wear (line 128). Were the participants were sleeping with the devices on? Please clarify. Normally, 10 hours is considered the cut-off for minimum wear time to equal a valid day.

Reply: Non-wear time is usually set at 60 minutes for hip-worn accelerometers. In our study, the accelerometers were worn at the wrist. There are currently no consistent recommendations for this location. A reduction of the non-wear time cut-off may result in an underestimation of sedentary behaviour. We only analysed physical activity and not sedentary behaviour. Therefore, non-wear time set at 30 minutes does not influence the amount of physical activity but rather reduces the risk of type II errors (Knaier 2019). 

The participants were asked to wear the accelerometers day and night (24 h). This information was added in line 125. Using the 20 h cut-off only 4.8 % of the days were excluded from the analyses.

The authors state that the minimum was one day of wear time. How can you address intra-person variability in PA by climate with just 1 day of measurement?

Reply: The aim of this study was to examine the PA variability by weather rather than the intra-person variability of PA by weather. Therefore, we looked at PA on day-level and not on person-level. The median of accelerometer measurements per day was 13 (interquartile range: 8-17).

Please explain why activity level was classified based on mean daily CPM rather than CPM thresholds?

Reply: Accelerometer thresholds can lead to an under- or overestimation of MVPA as there is a wide variability in the physical functioning and fitness of older adults (Troiano, 2014, Rejeski et al. 2018). Moreover, from a technical view, the categorization of the data into intensities results in a loss of information. As our intention is to investigate total amount of PA, not PA intensities or adherence to PA guidelines, we choose to assess PA as mean daily CPM.

RESULTS:

Line 228 and in table 1: Include the mean # of valid days per person (with range) here and in table 2.

Reply: The distribution of valid days per person was added in lines 230-231 and table 2. Since table 1 is not person-based we assumed that the reviewer meant only table 2. If this is not the case, please let us know. 

Lines 244-247: Was the relationship between PA and age stronger in women or men - or no difference?

Reply: The negative relationship between PA and age was stronger in women (-36.4; 95%-CL: -44.4, -28.3) than in men (-21.2; 95%-CL: -28.6, -13.8), as stated in lines 247-249.

DISCUSSION:

Lines 311-314: What degree of substitution takes place? Do individuals engage in equal amounts of IPA when weather makes OPA untenable (or vice versa)? Or is there magnitude of change? (I.e. said individual engages in half the IPA that they would have if weather was nicer and they went outside?)

Reply: We added a figure to the supplementary material to give more insight into the inverse relationship between OPA/ IPA and cloud cover (see S1 Fig). The text was changed to clarify that this is only one possible explanation (see lines 314-316). 

References

Knaier R, Höchsmann C, Infanger D, Hinrichs T, Schmidt-Trucksäss A. Validation of automatic wear-time detection alorithms in a free-living setting of wrist-worn and hip-worn ActiGraph GT3X+. BMC Public Health. 2019;19:244.

Rejeski WJ, Walkup MP, Fielding RA, King AC, Manini T, Marsh AP, McDermott M, Miller EY, Newman AB, Tudor-Locke C, Axtell RS, Miller ME. Evaluating accelerometry thresholds for detecting changes in levels of moderate physical activity and resulting major mobility disability. J Gerontol A Biol Sci Med Sci. 2018;73(5):660-667.

Troiano RP, McClain JJ, Brychta RJ et al. Evolution of accelerometer methods for physical activity research. Brit J Sports Med. 2014; 48:1019–1023.

---

## [Editor Report · Decision Letter 1]

7 Jan 2020

Accelerometer-assessed outdoor physical activity is associated with meteorological conditions among older adults: cross-sectional results from the OUTDOOR ACTIVE study

PONE-D-19-20850R1

Dear Dr. Albrecht,

We are pleased to inform you that your manuscript has been judged scientifically suitable for publication and will be formally accepted for publication once it complies with all outstanding technical requirements.

With kind regards,

Joe Robert Nocera

Academic Editor

PLOS ONE

---

## [Editor Report · Acceptance letter]

14 Jan 2020

PONE-D-19-20850R1 

Accelerometer-assessed outdoor physical activity is associated with meteorological conditions among older adults: cross-sectional results from the OUTDOOR ACTIVE study 

Dear Dr. Albrecht:

I am pleased to inform you that your manuscript has been deemed suitable for publication in PLOS ONE. Congratulations! Your manuscript is now with our production department. 

With kind regards,

on behalf of

Dr. Joe Robert Nocera 

Academic Editor

PLOS ONE